# Effects of microbial-derived long-chain polyunsaturated fatty acids from *Aurantiochytrium limacinum* BCC52274 and *Mortierella* sp. on growth and immunity in *Litopenaeus vannamei* post-larvae

Virak Visudtiphole[1], Panida Unagul[2], Sage Chaiyapechara[3],
Waraporn Jangsutthivorawat[3], Metavee Phromson[3], Siriporn Tala[3],
Pacharawan Deenarn[1], Punsa Tobwor[1], Pisut Yotbuntueng[1], Surasak Jiemsup[1],
Suganya Yongkiettrakul[1], Looksorn Koichai[1], Wananit Wimuttisuk[1]*

1 Biosensing and Bioprospecting Technology Research Group, National Center for Genetic Engineering and Biotechnology, National Science and Technology Development Agency, Pathum Thani, Thailand,
2 National Biobank of Thailand, National Science and Technology Development Agency, Pathum Thani, Thailand, 3 Integrative Aquaculture Biotechnology Research Group, National Center for Genetic Engineering and Biotechnology, National Science and Technology Development Agency, Pathum Thani, Thailand

* wananit.wim@biotec.or.th

## Abstract

The rapid growth of the aquaculture industry has increased the demand for feed ingredients, resulting in a shortage of fishmeal and fish oil, the latter of which serves as a source of essential fatty acids in aquaculture feed. As dietary supplementation of long-chain polyunsaturated fatty acids (LC-PUFA) improved growth and strengthened immunity in marine larvae, alternative sources for LC-PUFA are required to maintain sustainable aquaculture practice. This study explored the use of *Aurantiochytrium limacinum* BCC52274 (AL) and oil extracted from *Mortierella* sp. (ARASCO) as the source of LC-PUFA for the Pacific white shrimp *Litopenaeus vannamei* post-larvae (PLs) by using *Artemia* as a carrier. The *Artemia* were first enriched with combinations of AL and ARASCO with varying ratios of DHA:ARA at 100:0, 75:25, 50:50, 25:75, and 0:100, which were designated as Groups A, B, C, D, and E, respectively. The unenriched *Artemia* in Group R served as a control. The *L. vannamei* PL1 were fed with these *Artemia* for 18 days, revealing that the PL18 in Group A contained the highest biomass and average body weight among all feed groups. Meanwhile, other ratios of DHA:ARA supplementation were more beneficial to shrimp immunity, as the PL18 from Group D expressed the highest levels of *prophenoloxidase II* transcripts among all feed groups. The increasing proportion of ARA in the enriched *Artemia* also increased the levels of anti-inflammatory eicosanoids, including 15-deoxy-$\Delta^{12,14}$-prostaglandin J$_2$, 11-hydroxyeicosatetraenoic acid (11-HETE), and 12-HETE. When these PLs were infected with white spot syndrome virus (WSSV), the PLs from

**Data availability statement:** All relevant data are within the manuscript and its Supporting Information files.

**Funding:** This research has received funding support from the Functional Ingredient & Food Innovation Program, National Center for Genetic Engineering and Biotechnology [grant number P19-51807] to VV and the NSRF via the Program Management Unit for Human Resources & Institutional Development, Research and Innovation [grant number B05F640184] to WW.

**Competing interests:** The authors have declared that no competing interests exist.

Groups A, C, D, and E contained lower WSSV copy numbers compared with Group R, suggesting that the supplementation of LC-PUFAs from AL and ARASCO strengthened the immunity of these PLs against viral infection. However, the challenge with *Vibrio harveyi* resulted in no significant difference in the mean survival rates of PLs in all feed groups. Our results indicate that AL and ARASCO are more sustainable alternative sources of essential fatty acids that can be used strategically to enhance the growth and immunity of *L. vannamei* PLs.

## Introduction

Dietary supplementation of long-chain polyunsaturated fatty acids (LC-PUFA) enhances the growth and health of larval marine animals in domestication [1–6]. The expansion of the aquaculture industry has increased the demand for LC-PUFA supplementation, which has traditionally been sourced from fish oil. Due to the scarcity of fish oil, alternative sources of essential fatty acids are required for future feed production [7,8]. *Aurantiochytrium limacinum* (AL), which produces high levels of n-3 LC-PUFA, has previously been shown to promote growth, swimming strength, and hypo-salinity tolerance in the Pacific white shrimp *Litopenaeus vannamei* post-larvae (PLs) [9]. Similarly, *Mortierella alpina* can produce lipids mainly composed of triacyl-glycerol with high quantities of arachidonic acid (ARA; 20:4n-6) [10,11] and used as feed supplementation in poultry [12].

In shrimp post-larvae (PLs), LC-PUFA supplementation was often performed via the enrichment of *Artemia* live feed [4,13]. *Artemia* enriched with n-3 highly unsaturated fatty acids has been shown to enhance resistance to osmotic stress in the PLs of black tiger shrimp *Penaeus monodon* [13]. Similarly, *Artemia* supplemented with fish liver oil rich in docosahexaenoic acid (DHA, C22:6n-3) promoted the growth and health quality of the Indian prawn *Penaeus indicus* PLs [4]. Although the growth-promoting effects of n-3 LC-PUFA in feed supplementation were confirmed, the effects of dietary arachidonic acid (ARA, C20:4n-6) on PL growth varied in the juvenile freshwater prawn *Macrobrachium rosenbergii* and the Chinese prawn broodstock *Penaeus chinensis* [14,15], but showed positive correlations in *P. monodon* [16,17]. ARA supplementation also positively affects *L. vannamei* immune responses [18,19]. The mechanism by which ARA enhances immune responses is likely through the production of eicosanoids, a series of oxygenated polyunsaturated fatty acids that serve as signaling molecules for the pro- and anti-inflammatory responses. Although the roles of eicosanoids in immunity have been established in mammals [20,21], the known functions of eicosanoids in crustaceans are limited to ovarian maturation and sperm development [22–27]. The correlations between dietary LC-PUFA and the levels of eicosanoids have yet to be demonstrated in *L. vannamei*.

Earlier, n-3 LC-PUFA supplementation was studied in *L. vannamei* PLs and found to promote growth performance, swimming strength, and hypo-salinity tolerance in the PLs. However, it did not affect the immunity and disease resistance of the shrimp [9]. On the other hand, n-6 LC-PUFAs are more related to the primary inflammatory

mechanism defending against diseases [28]. In this study, dietary supplementation of *Aurantiochytrium limacinum* BCC52274 (AL) and *Mortierella*-extracted oil (ARASCO) as n-3 and n-6 LC-PUFA sources for *L. vannamei* PL was investigated. The supplementation was performed by enriching live *Artemia*, which was later fed to the PLs. The growth performance and changes in the free fatty acid profile of the PLs were examined. Additionally, a connection between n-3 and n-6 LC-PUFA supplementation and the levels of eicosanoids was established for the first time in *L. vannamei*. Immunological effects of the n-3 and n-6 LC-PUFA supplementation were assessed using the transcriptional analysis of shrimp immune genes as well as pathogenic challenge. Our analysis provided information for a strategic selection of microbial-derived LC-PUFA supplementation for *L. vannamei* PL to suit the different requirements of the aquaculture industry.

## Materials and methods

### The preparation of *A. limacinum* BCC52274

AL strain BCC52274 (obtained from the National Biobank of Thailand) was cultured in a medium composed of 1.5% (w/v) sea salt, 10% (w/v) glucose, and 2% yeast extract. The culture was maintained in a 10-L fermenter at 28°C with aeration for 3 days as previously described [29,30]. AL cells were harvested by 3,000×g centrifugation and washed twice with 0.9% normal saline. Cells were freeze-dried, sealed in light-protected plastic bags, stored at −20°C, and used within 1 month.

### The formulation of the enrichment emulsion

ARASCO (DSM, Switzerland), an ARA-rich oil extracted from *Mortierella* sp., was used as the dietary n-6 LC-PUFA source. The oil-in-water emulsion was prepared using a vacuum blender by blending ARASCO with 30-ppt seawater. Sodium polysorbate (12 μL/L) and alpha-tocopherol (4 mg/L) were added as an emulsifier and antioxidant, respectively [31,32]. Dried AL cells were dispersed into the emulsion through a 120T polyester mesh (100-μm pore size). Proportions of the dried AL and ARASCO contents in the cell and emulsion mixture varied according to the DHA and ARA ratios in the S1 Table. The control group (R, 0:0) contained only sodium polysorbate and alpha-tocopherol without AL and ARASCO.

### The enrichment of live *Artemia*

*Artemia* cysts (Salt Lake, USA) were hatched in 30 ppt seawater with vigorous aeration for 27–31 hours to reach the instar-II stage. *Artemia* was then enriched with the prepared emulsion for 4 hours at the ratio of 200,000 individuals in 1 L of 30-ppt water containing 90 mg of the combined total DHA and ARA content. Finally, the *Artemia* were harvested, stored at 4°C, and used for the experiment within 24 hours.

### Experimental animals

Specific pathogen-free *L. vannamei* at stage-3 mysis were obtained from a commercial hatchery in Eastern Thailand. The mysis were randomly distributed into each rearing tank (cylindrical cone-bottom fiberglass tank, 0.75 m diameter), which contained 150 L of 30-ppt sea water at the density of 100 individuals/L with aeration supplied by air stones. The mysis was acclimated until > 80% of the shrimp entered the post-larval (PL)-1 stage (approximately 1.5 days) before the actual feeding experiment started. During acclimatization, instar-I *Artemia* knocked by quick blanching was offered to the larvae for 6 meals/day. All animal experiment protocols were approved by the National Center for Genetic Engineering and Biotechnology Institutional Animal Care and Use Committee (Approval code: BT-Animal 08/2563) and carried out according to the relevant guidelines and regulations.

### Feeding experiment

The experiment was divided into six groups, each randomly assigned to 3 rearing tanks. Fifteen thousand shrimp were randomly distributed into each tank, containing 150 L of 30-ppt water. At PL13, the water volume was adjusted to 180 L to

accommodate the larger size of the PL. The enrichment experiment began with the PL-1 stage, which were fed 6 times daily with *Artemia* enriched with varying DHA:ARA ratios for 18 days. Each PL group was fed to satiation, and the feeding ration for the next meal was adjusted based on the leftover amount of *Artemia* to compensate for the potential difference in the quantity of *Artemia* consumed caused by the varying enrichment.

To maintain water quality, daily water exchange was conducted at the rate of 30–50%. The water quality was maintained in the following ranges: dissolved oxygen level > 5 mg/L, pH between 7.5–8.5, temperature between 30–32°C, salinity at 30 ppt, ammonia nitrogen < 0.03 mg/L, nitrite nitrogen < 1 mg/L, and alkalinity between 100–150 mg $CaCO_3$/L.

### Evaluation of growth performance and sample collection

On day 18, the PL18 were starved for 5 hours before being collected to measure the growth performance. The average body weight was determined by counting the number of PL ranging from 1.5–2 mg total weight (2 count replicates/tank). The survival rate was calculated from the final biomass and the average individual weight data. The length of PL from each tank (N = 100) was measured from the end of the rostrum to the end of the telson. The percentage coefficient of length variation was calculated from the average length of the PL.

$$\text{Percent coefficient of length variation} = \frac{\text{Mean}}{(\text{Standard deviation})} \times 100$$

### Pathogen challenges

The *Vibrio harveyi* challenge was performed by transferring 20 PLs from each feeding tank to a glass cylinder containing 25 L of aerated 30-ppt water with the *V. harveyi* at 0, $7.96 \times 10^6$, or $1.58 \times 10^7$ CFU/mL. After 24 hours, the water volume in each cylinder was adjusted to 30 L with non-contaminated 30-ppt water, in which 50% of the water was exchanged daily, and the water was always aerated. During the challenge, the PL19 were fed with plain instar-II *Artemia* and maintained for 2 days before assessment of the survival.

The white spot syndrome virus (WSSV) infection in PLs was induced by ingesting the muscle tissue homogenate from WSSV-infected *L. vannamei* juveniles. The infected meat was prepared by injecting 300 µL of WSSV at a concentration of $10^2$ copies/µL into 30−40 g juveniles (7 months old). The infected juveniles were maintained in 25-ppt water for 3 days. The muscle tissue of the infected moribund shrimp was dissected, minced, and stored at −20°C until offered to the challenged PL.

For the WSSV challenge, 35 PLs from each feeding experiment tank were transferred to a glass cylinder containing 20 L of aerated 30-ppt water. The PLs were starved for 18 hours and then fed twice within 6 hours with the prepared WSSV-infected meat at the rate of 66.29% of their average body weight/meal. A control was also set up separately for each group, in which PLs were fed with muscle tissue homogenate from a non-infected juvenile. The PLs were maintained by feeding on non-enriched Instar-II *Artemia*. Fifty percent of the rearing water was changed daily. After two days, the challenged PLs were collected, flash-frozen in liquid $N_2$ and stored at −80°C.

### Quantitative real-time PCR analysis to detect WSSV copy number

The DNA of WSSV-infected PLs (N = 35) was extracted using the QiaAmp DNA Mini Kit (Qiagen, Hilden, Germany) and treated with RNase A (Qiagen). The PCR was performed according to Srisala et al. 2008, in which the WSSV-detection primers are WSSV-229F1: 5' GATGGAAACGGTAACGAATCTGAA 3' and WSSV-447R1: 5' CAGAGCCTAGTCTAT-CAATCA T 3' [33]. The DNA template used to construct the standard curve was a purified 1447-bp amplicon from the WSSV-DNA [34]. The standard curve was constructed using a 10-fold serial dilution of the WSSV-DNA from $10^2$ to $10^7$ copies/reaction, resulting in a 99.4% amplification efficiency. The qPCR reaction included 0.1 ng of the extracted DNA, 10

 

μL of the 2x SYBR Green (Biorad, California, USA), 5 μM of each primer, and PCR-grade water for a final volume of 20 μL. The PCR condition was 1 cycle of 98°C for 3 min, 40 cycles of 98°C for 30 s, 55°C for 30 s, and 72°C for 30 s, with 1 final cycle of 72°C for 45 s.

### Fatty acid analysis using gas chromatography flame ionization detector (GC-FID)

Fatty acid extraction and trans-esterification were performed according to Visudthiphole et al. (2018) [9]. Briefly, the fatty acids from the freeze-dried samples were directly extracted and trans-esterified in 4% sulfuric acid in methanol by heating to 90°C for 1 hour [35]. Heptadecaenoic acid (C17:0) or nonadecanoic acid (C19:0) (Sigma-Aldrich) was used as an internal standard. The extracted fatty acid samples were analyzed using the Shimadzu GC-17A high precision gas chromatography (Shimadzu, Japan) equipped with the Omegawax™ 250, 30 m x 0.25 mm fused silica capillary column (Supelco, Sigma-Aldrich, Massachusetts, USA) and flame ionization detector (FID) following the analysis performed by Visudtiphole et al. 2021 [30]. Helium was selected as a carrier gas with an average linear velocity of 30 cm/s. Temperatures for the injector and detector were set at 250°C and 260°C, respectively. The initial column temperature was set and maintained at 200°C for 10 min and then increased at a rate of 20°C/min to 230°C. Subsequently, the column temperature was held for 17 min. The peak identification and quantification were analyzed based on the retention times relative to fatty acid methyl ester standards (Supelco 18919−1 AMP, Sigma-Aldrich). Each fatty acid was identified and quantified based on the retention time of each peak relative to the fatty acid methyl ester standard peaks.

### RNA extraction and cDNA synthesis

*L. vannamei PL18* ($N = 30$) were subjected to total RNA extraction using the Trizol reagent (Invitrogen, California, USA). The obtained RNA samples were treated with RNase-Free DNase (Promega, Wisconsin, USA). One microgram of total RNA was reverse transcribed using the RevertAid™ First Strand cDNA Synthesis Kit with oligo $(dT)_{18}$ primers (Thermo Scientific, Massachusetts, USA). The reaction volume was 20 μL per reaction, resulting in a final total RNA concentration of 50 ng/μL. The cDNA concentration from each sample was determined using the Thermo Scientific™ NanoDrop™ One Microvolume UV-Vis Spectrophotometer (Thermo Scientific).

### Gene expression analysis

Expression analysis of shrimp immune genes was performed on cDNA samples using the SsoFast EvaGreen Supermix (Bio-Rad, California, USA) according to the manufacturer's instructions. One hundred nanograms of cDNA were used as templates for each real-time PCR reaction. Primer sequences for immune genes are provided in the S2 Table. Melt curve analysis was performed for each pair of primers as previously described [36]. Amounts of target gene transcripts relative to *elongation factor 1α* (*EF1α*) were obtained using the standard curve method. Amplification efficiency was between 95–105%. The specificity of the PCR product was confirmed using melting curve analysis with a continuous fluorescence reading at 0.5°C increments from 55°C to 95°C.

### Ethyl acetate extraction of PL samples

Ethyl acetate extraction of PL was adapted from Yotbuntueng et al. 2022 [27]. PLs were homogenized in liquid nitrogen and diluted in HBSS (Sigma-Aldrich) at a concentration of 0.1 g/mL. The homogenates were mixed with 10% BHT in HPLC-grade ethanol (w/v) and 5 μL of 20 μM $PGE_2$-$d_4$, 5-HETE-$d_8$, and EPA-$d_5$ as an antioxidant and internal standards, respectively. An equal volume of ethyl acetate was added to the PL homogenates, and the mixture was mixed vigorously for 15 min. To collect the organic phase, the extract was spun down at 8,228 × g for 10 min at 20°C. Another equal volume of ethyl acetate was added to the aqueous phase, and the extraction process was repeated. The PL extract was dried using the rotary evaporator and stored at −80°C.

## Ultra-high performance liquid chromatography-high resolution tandem mass spectrometry (UHPLC-HRMS/MS) analysis

The UHPLC-HRMS/MS analysis was performed via a DIONEX 3000 RS UHPLC system coupled with an Orbitrap Fusion™ Tribrid™ mass spectrometer (Thermo Fisher Scientific). The system was operated using the Xcalibur software (version 4.4.16.14) and calibrated with the Pierce™ FlexMix™ calibration solution (Thermo Fisher Scientific). Parameters for the UHPLC conditions and mass spectrometer analysis were conducted as previously described [27].

### Statistical analysis

The statistical analyses were performed using a one-way analysis of variance (ANOVA) with Duncan's multiple range test (DMRT) (SPSS 11.5.0) with a threshold for significance at $P < 0.05$. Dunnett's test was performed for the WSSV and *V. harveyi* challenges to compare the LC-PUFA supplemented samples to the control group (* for $P < 0.05$ and ** for $P < 0.01$).

## Results

The effects of supplementing n-3 and n-6 LC-PUFAs derived from AL and ARASCO in *L. vannamei* PLs were determined according to the experimental outline shown in Fig 1. Briefly, *Artemia*, a live feed commonly used for PL, were enriched with different ratios of AL to ARASCO via immersion. Five groups of the enriched *Artemia* (Groups A to E) and a control (Group R) were fed to *L. vannamei* PL1 for 18 days. The effects of n-3 and n-6 LC-PUFA enrichment were performed by monitoring biomass, fatty acid profiles, eicosanoid profiles, and gene expression levels. The remaining PLs were divided into two groups to be challenged with WSSV and *V. harveyi* to determine the immunological effects of n-3 and n-6 LC-PUFA supplementation on the PLs.

GC-FID analysis was performed on the *Artemia,* which had previously been enriched with AL and ARASCO to obtain different ratios of DHA:ARA. The fatty acid profiles of *Artemia* with and without supplementation are shown in S3 Table, in which the ratios of DHA:ARA calculated from the percentage of total fatty acids (%TFA) were 13.95:1, 1.19:1, 0.18:1, 0.10:1, and 0.00:1 for the *Artemia* in Groups A, B, C, D, and E, respectively.

The supplemented *Artemia* were given to *L. vannamei* PL1 for 18 days, and the resulting PL18 were analyzed using GC-FID to determine fatty acid profiles (Table 1). The fatty acid compositions and the total fatty acid content in PL18 corresponded with those in the offered feed. For example, the short-chain fatty acid (SFA) content of the PLs from Group A was the highest among all PL groups due to the high content of C16:0 found in the AL. Similarly, the PLs fed with *Artemia* enriched with increasing proportions of AL to ARASCO also contained increasing levels of n-3 LC-PUFA due to the high levels of DHA (C22:6) and DPA (C22:5) in AL. On the other hand, the PL18 fed with a higher proportion of ARASCO contained higher levels of n-6 LC-PUFA, which contributed solely from the high content of ARA (C20:4). Moreover, the ratios of DHA:ARA in the supplemented PLs mimicked those found in the offered feed, which gradually decreased from Groups A to E.

### Growth performance analysis of PL with LC-PUFA supplementation

The supplementation of DHA:ARA at 100:0, 75:25, and 50:50 ratios was beneficial to the PL growth as the PL18 from Groups A, B, and C displayed higher biomass and body length than Group R (the control group) (Figs 2A and 2B). Similarly, the PLs from Groups A and B showed higher wet and dried body weight than Group R (Figs 2C and 2D), further confirming that DHA supplementation was beneficial to the PL growth. However, the increasing proportion of ARA supplementation did not increase the PL body weight. All PL groups with LC-PUFA supplementation showed reduced feed conversion rates (FCR) when compared to Group R (Fig 2E), suggesting that any combination of n-3 and n-6 LC-PUFA supplementation was beneficial to the PL growth efficiency. Nevertheless, the survival rate and the percent coefficient of size variation were unaffected by LC-PUFA supplementation (Figs 2F and 2G).

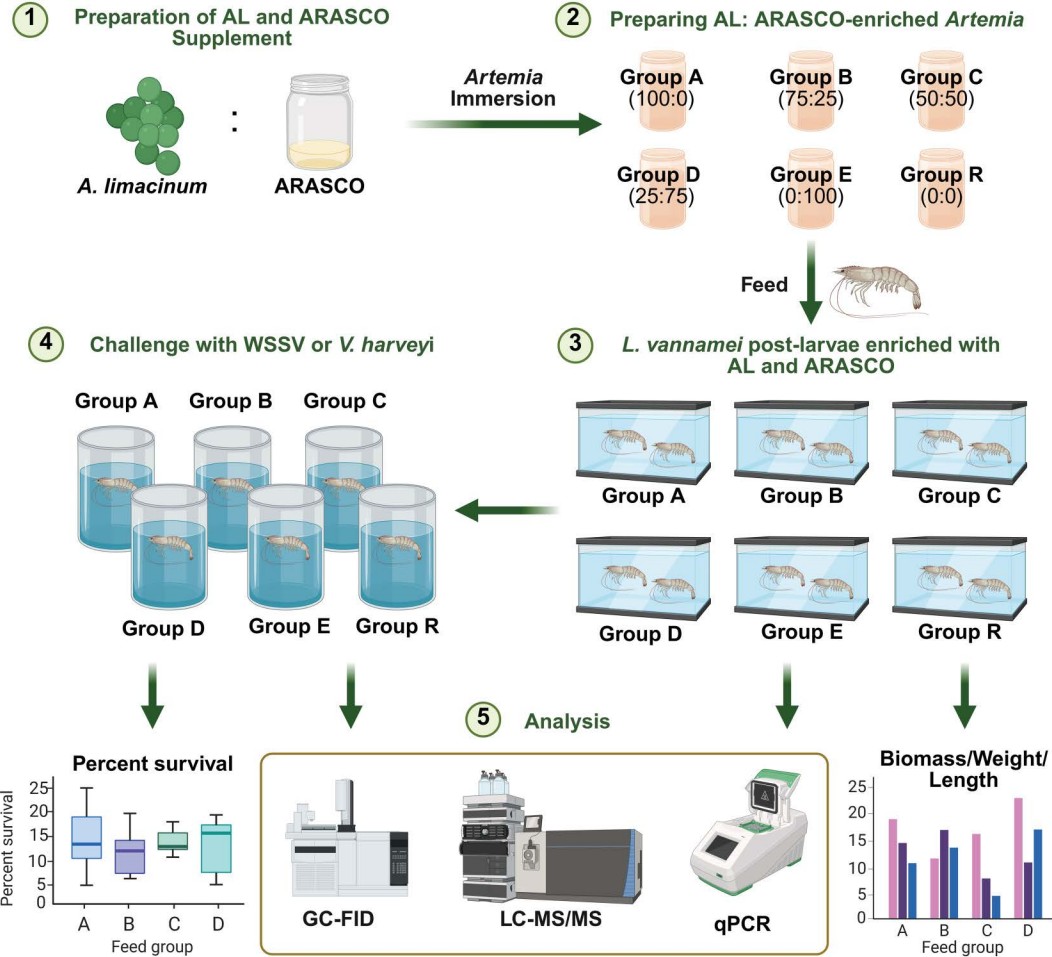

**Fig 1. Schematic diagram outlining the experiments used to determine the effects of n-3 and n-6 LC-PUFA from AL and ARASCO on *L. vannamei* PLs.** Created in BioRender. Wimuttisuk, W. (2025) https://BioRender.com/nknnhk8.

## Transcriptional analysis of shrimp immune genes

To determine whether the increasing levels of LC-PUFA in PLs also affect the transcriptional levels of immune genes, the qPCR analysis was performed to monitor the transcription levels of *prophenoloxidase I* (*ProPO-I*), *prophenoloxidase II* (*ProPO-II*), *prophenoloxidase-activating enzyme* (*ppA*), *penaeidin 3a* (*PEN3a*), and *superoxide dismutase* (*SOD*) in PLs. The transcription levels of *ProPO-I*, *ppA*, *PEN3a*, and *SOD* were unaffected by the LC-PUFA supplementation (Fig 3). However, the transcription levels of *ProPO-II* in Group D were the highest among all feed groups (Fig 3B), suggesting that the supplementation of DHA:ARA at a 25:75 ratio induced part of the shrimp immune system via the activation of the *ProPO-II* gene.

## Effects of LC-PUFA supplementation on the levels of eicosanoids in PLs

As both ARA and eicosapentaenoic acid (EPA; C20:5) served as eicosanoid precursors, the impact of LC-PUFA supplementation on eicosanoid profiles in PLs was monitored using the UHPLC-HRMS/MS analysis. Nine eicosanoids, including two prostaglandins, three isomers of hydroxyeicosatetraenoic acids (HETEs), and four isomers of

**Table 1. Fatty acid compositions and total fatty acid (TFA) contents (% dry weight) of PL1 shrimp before the experiment and PL18 supplemented with varying ratios of DHA:ARA.**

| Fatty acid | PL1 | PL18 | | | | | |
|---|---|---|---|---|---|---|---|
| | | R | A | B | C | D | E |
| C14:0 | 0.014 | 0.013±0.000[a] | 0.022±0.001[b] | 0.017±0.001[b] | 0.016±0.001[a,b] | 0.015±0.000[a,b] | 0.015±0.001[a,b] |
| C15:0 | 0.052 | 0.048±0.004[a] | 0.061±0.001[b] | 0.059±0.000[b] | 0.072±0.001[c] | 0.074±0.000[c] | 0.074±0.002[c] |
| C16:0 | 0.608 | 0.684±0.007[b] | 0.963±0.021[b] | 0.699±0.013[b] | 0.668±0.027[b] | 0.584±0.024[a] | 0.591±0.007[a] |
| C18:0 | 0.495 | 0.475±0.006[a] | 0.530±0.019[a,b] | 0.493±0.025[a] | 0.601±0.016[c] | 0.606±0.008[c] | 0.586±0.006[b,c] |
| C22:0 | 0.017 | 0.000±0.000[a] | 0.025±0.001[bc] | 0.025±0.001[c] | 0.021±0.001[b] | 0.0213±0.001[b] | 0.020±0.001[b] |
| ∑SFA | 1.186 | 1.220±0.014[a] | 1.602±0.034[c] | 1.292±0.028[a] | 1.379±0.038[b] | 1.301±0.032[a,b] | 1.287±0.006[a] |
| C16:1 | 0.245 | 0.276±0.008[b] | 0.216±0.009[a] | 0.194±0.007[a] | 0.199±0.008[a] | 0.197±0.008[a] | 0.204±0.008[a] |
| C18:1 | 0.805 | 0.842±0.011[b] | 0.742±0.008[b] | 0.718±0.015[b] | 0.691±0.031[b] | 0.723±0.035[b] | 0.761±0.009[b] |
| C20:1 | 0.093 | 0.115±0.002[a,b] | 0.106±0.003[a,b] | 0.125±0.006[b] | 0.106±0.003[a] | 0.105±0.002[a] | 0.105±0.006[a] |
| ∑MUFA | 1.143 | 1.233±0.011[b] | 1.063±0.016[a] | 1.037±0.024[a] | 0.996±0.042[a] | 1.026±0.044[a] | 1.07±0.005[a] |
| C18:3 | 0.926 | 1.196±0.040[b] | 0.855±0.010[a] | 0.909±0.035[a] | 0.805±0.048[a] | 0.847±0.055[a] | 0.935±0.028[a] |
| C20:5 | 0.481 | 0.315±0.002[b] | 0.322±0.003[b] | 0.298±0.002[a] | 0.287±0.005[a] | 0.293±0.008[a] | 0.293±0.001[a] |
| C22:5 | 0.014 | 0.000±0.000[a] | 0.097±0.002[d] | 0.040±0.001[c] | 0.038±0.001[c] | 0.015±0.001[b] | 0.000±0.000[a] |
| C22:6 | 0.191 | 0.029±0.006[a] | 0.54±0.012[d] | 0.252±0.002[c] | 0.255±0.003[c] | 0.105±0.004[b] | 0.042±0.001[a] |
| ∑n-3 | 1.613 | 1.540±0.047[c] | 1.815±0.024[d] | 1.499±0.034[b,c] | 1.386±0.052[a,b] | 1.260±0.058[a] | 1.270±0.028[a] |
| C18:2 | 0.293 | 0.334±0.007[c] | 0.265±0.003[a,b] | 0.264±0.009[a,b] | 0.244±0.010[a] | 0.264±0.011[a,b] | 0.286±0.002[b] |
| C20:2 | 0.040 | 0.191±0.008[a,b] | 0.188±0.006[a,b] | 0.208±0.005[b] | 0.190±0.008[a,b] | 0.189±0.017[a,b] | 0.160±0.009[a] |
| C20:3 | 0.008 | 0.006±0.001[a] | 0.007±0.000[a] | 0.006±0.000[a] | 0.008±0.001[a] | 0.070±0.000[a] | 0.017±0.001[a] |
| C20:4 | 0.206 | 0.109±0.001[a] | 0.157±0.001[c] | 0.141±0.001[b] | 0.200±0.001[d] | 0.217±0.002[e] | 0.255±0.002[f] |
| ∑n-6 | 0.547 | 0.640±0.016[a,b] | 0.616±0.003[a] | 0.618±0.012[a] | 0.642±0.015[a,b] | 0.676±0.023[b,c] | 0.718±0.005[c] |
| ∑PUFA | 2.160 | 2.180±0.061[c] | 2.431±0.022[d] | 2.117±0.045[b,c] | 2.027±0.066[a,b,c] | 1.937±0.057[a] | 1.988±0.024[a,b] |
| ∑HUFA | 0.941 | 0.650±0.014[a] | 1.442±0.007[e] | 1.094±0.013[d] | 1.106±0.014[d] | 0.953±0.016[c] | 0.893±0.010[b] |
| n-3/n-6 | 2.946 | 2.407±0.027[c] | 2.944±0.052[d] | 2.423±0.029[c] | 2.160±00.043[b] | 1.863±0.124[a] | 1.769±0.051[a] |
| DHA/ARA | 0.926 | 0.267±0.051[a] | 3.445±0.069[e] | 1.790±0.025[d] | 1.276±0.022[c] | 0.486±0.024[b] | 0.164±0.007[a] |
| ∑TFA | 4.490 | 4.633±0.083[b] | 5.097±0.064[c] | 4.446±0.087[a,b] | 4.401±0.142[a,b] | 4.263±0.128[a] | 4.345±0.032[a] |

hydroxyeicosapentaenoic acids (HEPEs), were detected in these PLs. Although ARA served as a precursor of prostaglandins and HETEs, the increasing proportion of ARA in the *Artemia* did not produce the same impact across all ARA-derived eicosanoids. For example, the levels of $PGF_{2\alpha}$ were comparable in all feed groups, while the levels of 15-deoxy-$\Delta^{12,14}$-prostaglandin $J_2$ (15d-$PGJ_2$) were at the lowest levels in Group R and steadily increased until it reached the highest levels in Group E (Figs 4A and 4B). Similarly, the increasing proportion of ARA in the LC-PUFA supplementation also impacted each HETE isomer differently. While the levels of 8-HETE were comparable among all feed groups (Fig 4C), the levels of 11-HETE and 12-HETE gradually increased as the proportion of ARA increased in the enriched *Artemia*, with the lowest levels of 11-HETE and 12-HETE in Group R and the highest levels in Group E (Figs 4D and 4E).

The GC-FID analysis of PL18 indicated that the levels of EPA in Groups A and R were higher than in the other feed groups (Table 1). Therefore, the levels of HEPEs, which are derivatives of EPA, would also increase in these PLs. However, the UHPLC-HRMS/MS analysis revealed that the increasing levels of EPA in PLs did not affect the production of HEPEs as the levels of 5-HEPE, 8-HEPE, 12-HEPE, and 18-HEPE were comparable among all feed groups (Figs 4F to 4I), suggesting that the increasing intake for EPA was utilized in other pathways to promote shrimp growth rather than for the biosynthesis of eicosanoids.

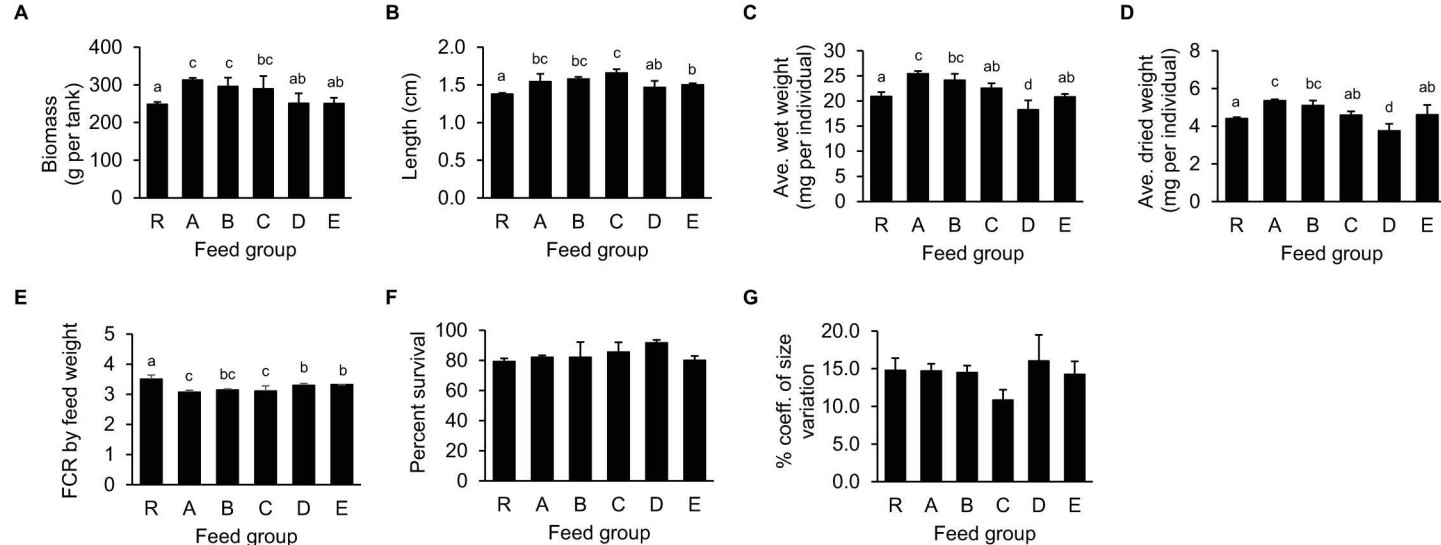

**Fig 2. Growth performance analysis of PL18 fed with *Artemia* enriched with graded ratios of DHA:ARA from AL and ARASCO, respectively.** PL18, which had previously been fed with *Artemia* containing different ratios of n-3 and n-6 LC-PUFAs, were examined for their (A) biomass in g per tank, (B) length, (C) average wet body weight, (D) average dried body weight, (E) FCR determined from the feed weight, (F) survival rate, and (G) percent coefficient of size variation. Error bars represent the standard deviation of the averaged data from the tank replicates (*N* = 3). Data were statistically analyzed using ANOVA with DMRT. Different letters above the bar graphs indicate significant differences between feed groups (*P* < 0.05).

### The effects of LC-PUFA supplementation on pathogenic challenges

To assess the effect of LC-PUFA supplementation on pathogenic resistance, PL18 samples from each feed group were transferred into glass tanks. After 24 hours of acclimation, the PL19 were infected with either WSSV or *V. harveyi* for 48 hours.

For WSSV infection, the PL19 were starved for 18 hours before being fed twice with either minced WSSV-free shrimp meat (control group) or the minced meat of WSSV-infected shrimp. The post-larvae were maintained for 48 hours, and the resulting PL22 were subjected to qPCR analysis to determine the WSSV copy number. The infected PLs from Groups A, C, D, and E contained lower WSSV copy numbers than those in Group R (Fig 5A), suggesting that LC-PUFA supplementation at 100:0, 50:50, 25:75, and 0:100 ratio of DHA:ARA is beneficial to shrimp immunity by reducing WSSV replication in the PLs. In particular, the PL22 in Group A and E showed the lowest numbers of WSSV replicates, suggesting that the supplementation with either DHA or ARA alone is more beneficial to the PLs during WSSV infection.

The *V. harveyi* challenge was performed by immersing 20 PL19 from each feed group with *V. harveyi* at 0, $7.96 \times 10^6$, or $1.58 \times 10^7$ CFU/mL for 24 hours to obtain control, low dose, and high dose of *V. harveyi* infection, respectively. The survival was assessed at 48 hours post-infection, revealing no significant difference in the mean survival rates of PLs in all feed groups due to considerable sample variation (Fig 5B). Therefore, LC-PUFA supplementation did not increase the PL survival rates during *V. harveyi* infection.

### Discussion

LC-PUFA supplementation has previously been shown to enhance the growth of PL in several marine organisms [1–6]. While the positive effect of DHA on growth is consistent in shrimp and most marine species [1–4], the results from ARA supplementation varied [5,6,17,37,38]. In the present study, dietary supplementation of LC-PUFA by *Artemia* enrichment was performed using different proportions of *A. limacinum* BCC52274 and ARASCO derived

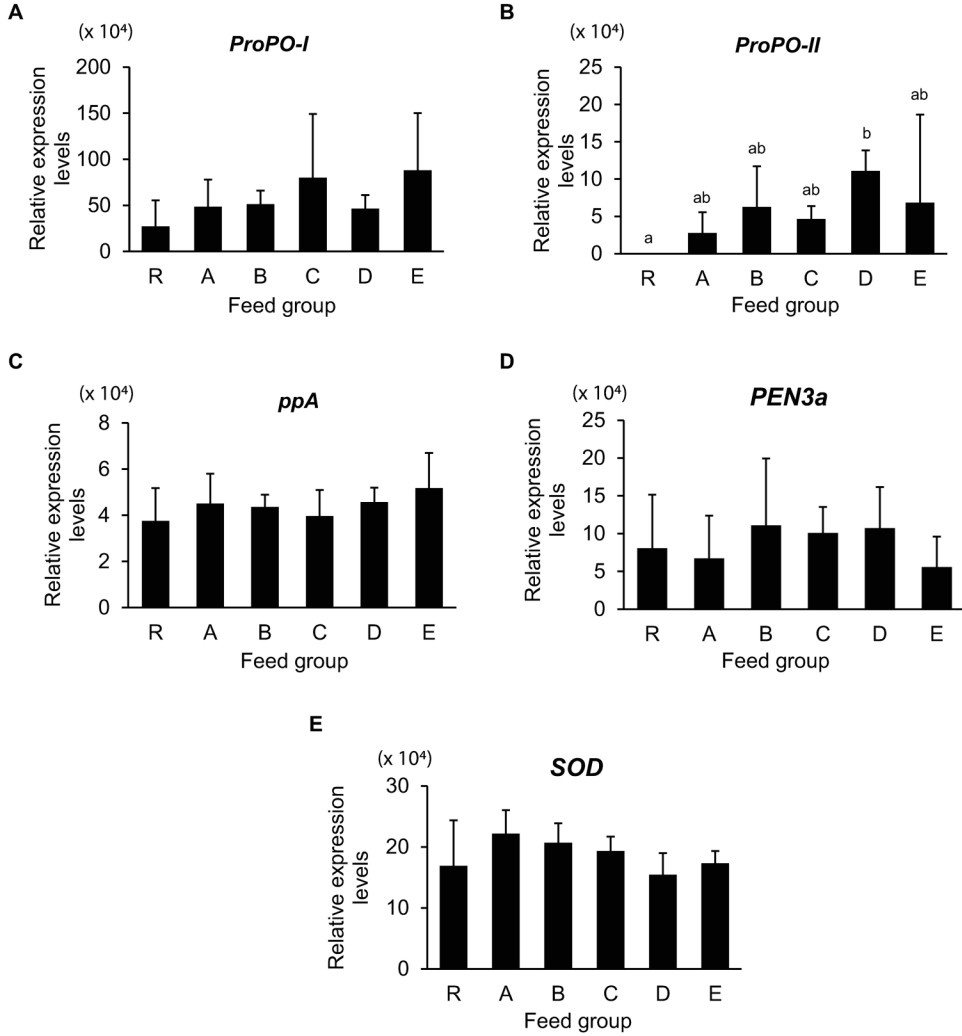

**Fig 3. Transcription levels of immune genes in PL18 fed with *Artemia* with varying ratios of n-3:n-6 LC-PUFA supplementation.** *L. vannamei* PL18 that has previously been fed with *Artemia* with no supplementation (Group R) and *Artemia* supplemented with n-3:n-6 at 100:0 (Group A), 75:25 (Group B), 50:50 (Group C), 25:75 (Group D), and 0:100 (Group E) ratios were homogenized under liquid $N_2$ and subjected to RNA extraction and cDNA synthesis. Expression levels of (A) *ProPO-I*, (B) *ProPO-II*, (C) *ppA,* (D) *PEN3a,* and (E) *SOD* relative to *EF1α* were determined using the qPCR analysis. The differences in transcription levels of shrimp immune genes among all feed groups were determined using ANOVA with DMRT and designated by different letters above the bar graphs ($P < 0.05$).

from *Mortierella* sp. to vary the ratios of DHA and ARA. Based on the data obtained in this study, we propose that the supplementation of microbial-derived PUFA can be selectively applied in different situations for the optimal *L. vannamei* post-larvae aquaculture production (Fig 6). During the pathogenic outbreak, the supplementation of DHA:ARA at a 25:75 ratio promoted the production of anti-inflammatory eicosanoids, including 15d-$PGJ_2$, 11-HETE, and 12-HETE, increased the transcription levels of an immune gene *ProPO-II*, and reduced the levels of WSSV at 24 hpi. As the prophenoloxidase system has been shown to protect shrimp against various pathogens, including *Enterocytozoon hepatopenaei* and *Vibrio parahaemolyticus* that caused acute hepatopancreatic necrosis disease (AHPND) [39,40], the supplementation of DHA:ARA at a 25:75 ratio during pathogenic outbreaks is recommended to promote shrimp immunity. However, as the energy obtained from ARA supplementation was allocated to heighten the immune

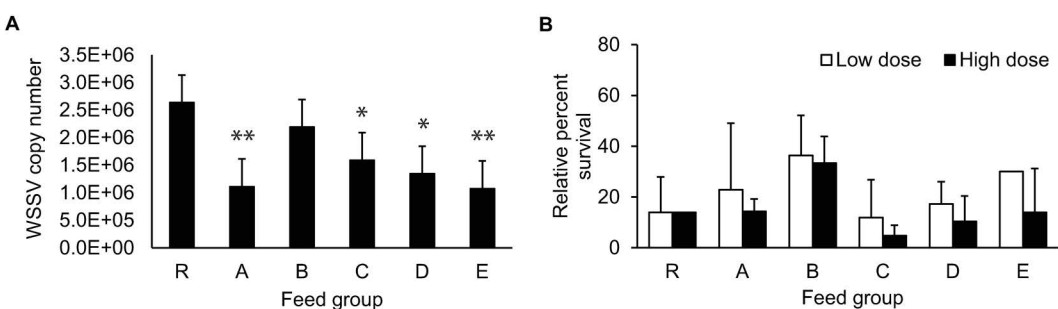

**Fig 4. The UPLC-HRMS/MS analysis of PL samples fed with enriched *Artemia*.** Levels of (A) PGF$_{2\alpha}$, (B) 15d-PGJ$_2$, (C) 8-HETE, (D) 11-HETE, (E) 12-HETE, (F) 5-HEPE, (G) 8-HEPE, (H) 12-HEPE, and (I) 18-HEPE from PLs in each feed group were quantified using the standard curve method. Error bars represent standard deviations, and different letters indicate statistically significant differences in the eicosanoid levels as determined by ANOVA using DMRT (P < 0.05).

**Fig 5. Effects of pathogenic infection in PLs fed with enriched *Artemia*.** PLs were challenged with (A) WSSV and (B) *V. harveyi*. Error bars represent the standard deviation of the averaged data from the tank replicates (N = 3). Statistically significant differences were determined by ANOVA using Dunnett's test to compare all LC-PUFA supplemented groups with the control group (* for P < 0.05 and ** for P < 0.01).

response, the biomass of these ARA-supplemented *L. vannamei* was comparable to that of the shrimp without PUFA supplementation.

During a low pathogenic outbreak, supplementation of DHA:ARA at 100:0 ratio would be the most beneficial to the shrimp aquaculture production as it resulted in the highest biomass, wet weight, and dry weight when compared to other feed groups. Moreover, shrimp supplemented with DHA:ARA at a 100:0 also contained the lowest WSSV copy number in PL, suggesting an additional benefit during WSSV outbreak.

The rapid expansion of the aquaculture industry resulted in the scarcity of feed ingredients [41,42]. In particular, the replacement of fish meal and fish oil, which are the primary sources of protein and essential fatty acids, respectively, is required for the sustainability of the aquaculture industry [7,8]. The supplementation of EPA and DHA from algal oil showed a significant improvement in specific growth rate, feed conversion rate, and survival in *L. vannamei* [43]. Similarly, the use of a fish oil-free dietary supplement with high levels of DHA (at 0.24% DHA) increased the growth of *L. vannamei* [44]. Several studies have examined the possibility of using fungal-derived LC-PUFA as fish oil replacement with varying outcomes [30,45,46]. The use of a different strain of *A. limacinum* has been shown to promote the growth and survival in *P. monodon* post-larvae [47]. Similarly, two studies by Yao et al. (2022) and Li et al. (2025) revealed that the supplementation of *Schizochytrium limacinum,* which is also known as *A. limacinum,* can improve growth performance and intestinal health of juvenile *L. vannamei* [48,49]. The use of *A. limacinum* BCC52274 has previously been tested on the growth of *Penaeus monodon* PLs [30]. The PLs fed with *Artemia* enriched with 100% *A. limacinum* showed the highest average body weight and biomass compared to other feed groups at 0, 50, and 75% enrichment. These data paralleled with the results obtained in this study, in which the PLs fed with *Artemia* supplemented with the highest ratio of DHA:ARA also showed the highest biomass and average wet and dry body weight among all feed groups. However, the effects of *A. limacinum* supplementation on the health and immunity of PLs were not investigated in *P. monodon*.

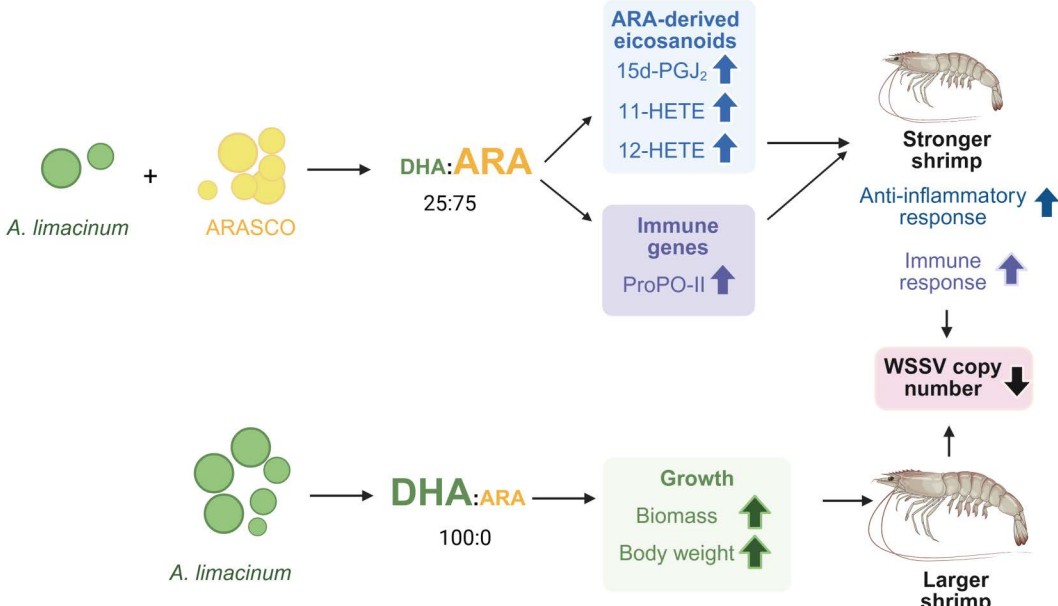

**Fig 6. A proposed model on the impact of DHA and ARA supplementation on *L. vannamei* health.** Created in BioRender. Wimuttisuk, W. (2025) https://BioRender.com/2bugr6t. Arrows pointing up indicate increasing levels of the designated metabolites, immune gene transcripts, biomass, body weight, or activation of specific pathways. An arrow pointing down indicates decreasing levels of WSSV copy number.

During the preparation of the enriched *Artemia*, the GC-FID analysis revealed that the ratios of DHA:ARA in the resulting enriched *Artemia* were lower than those of the prepared enrichment mixtures, particularly for Groups B, C, and D. The ratio changes commonly occur when a zooplankton is simultaneously enriched with different dietary substances [50–53]. This could have been attributed to several causes, including the different phases of AL and ARASCO in the mixture, different retro-conversion activities of DHA and ARA by *Artemia*, or the different stability of DHA and ARA present in the cell and oil droplet, respectively.

A study by Zhu et al. (2023) reported that dietary supplementation of n-6 LC-PUFA showed positive effects on growth, the antioxidative pathway, and the immune system in *L. vannamei* [6]. Moreover, dietary ARA and DHA supplementation increased the survival rates of *Apostichopus japonicus* and *L. vannamei* juveniles infected with *Vibrio sp.* [19,54]. These results are similar to the study by Nonwachai et al. 2010, in which the supplementation of ARA and DHA increased the *L. vannamei* survival from the *V. harveyi* infection more than individual supplementation of either ARA or DHA alone [19]. However, the supplementation of n-6 LC-PUFA derived from *Mortierella sp.* did not significantly impact the survival rate of *V. harveyi*-infected PLs in this study due to large sample variations among the replicates.

Although WSSV generally does not cause significant mortality in shrimp at the larval and PL stages [55,56], the WSSV challenge experiment was performed in this study to examine the capability of LC-PUFA supplementation to suppress the virus replication in its host. Free polyunsaturated fatty acids, including DHA and ARA, have been shown to suppress virus replication and infection [57–59]. However, this has not yet been evaluated in shrimp. In this study, dietary DHA and ARA were demonstrated for the first time to suppress replication of WSSV in the PLs. This is supported by a study by Hoffling et al. (2024), which demonstrated that *L. vannamei* fed with *Aurantiochytrium sp.* meal showed increased survival rates after the WSSV challenge [60]. Moreover, the dietary supplementation of ARA and linoleic acid (LA) has previously been correlated with increasing activities of SOD and catalase (CAT) in *L. vanmamei* [6]. Activities of lysozyme, acid phosphatase, and alkaline phosphatase also increased along with increasing amounts of dietary LA and ARA. However, the levels of *SOD* and *CAT* gene transcripts opposed the enzyme activity in shrimp that were supplemented with ARA [6]. As the supplementation of LC-PUFA from *A. limacinum* BCC52274 and ARASCO in this study also showed little impact on the transcription levels of shrimp immune genes, the effects of feed supplementation may be more prominent at the enzymatic levels rather than at the transcription levels.

Additional effects of ARA supplementation on the shrimp's immune system may stem from the increasing levels of eicosanoids in shrimp. ARA serves as a precursor in the eicosanoid biosynthesis pathway, which has been shown to regulate inflammation and immune response in insects and mammals [61–64]. In this study, the PLs fed with *Artemia* with decreasing ratios of DHA:ARA also showed increasing levels of 15d-PGJ$_2$, 11-HETE, and 12-HETE, suggesting that the amounts of dietary ARA were correlated with the levels of eicosanoids in *L. vannamei*. A study by Aguilar et al. (2012) revealed that increasing levels of ARA in the shrimp diet led to enhanced immune response to counteract the negative response of high-stocking density [18]. However, they concluded that the ARA-derived eicosanoids were not responsible due to the inconsistent levels of PGE$_2$ found in *L. vannamei*. Based on our results, however, we proposed that the eicosanoids responsible for enhancing the immune response are not PGE$_2$ but rather a different set of ARA-derived eicosanoids, such as 15d-PGJ$_2$, 11-HETE, and 12-HETE. Additionally, the ethanol extract of *A. limacinum* has anti-inflammatory properties by reducing the expression of proinflammatory cytokine genes, including tumor necrosis factor α, interleukin-1β, and interleukin-6 when tested *in vitro* using RAW264 murine macrophage cells [65].

## Conclusions

The use of *A. limacinum* BCC52274 and oil extracted from *Mortierella sp.* (ARASCO) can serve as an alternative source of LC-PUFA to promote the growth and immunity of *L. vannamei* PL by using *Artemia* as a carrier. The PL supplemented with DHA:ARA at a 100:0 ratio resulted in the highest biomass and average body weight among all feed groups. On the other hand, the supplementation of DHA:ARA at a 25:75 ratio resulted in the highest induction of an immune gene,

*prophenoloxidase II*. The increasing proportion of ARA in the supplements was also positively correlated with the levels of anti-inflammatory eicosanoids, making it suitable for feed supplementation during pathogenic outbreaks. Our results provide supporting evidence that the supplementation of LC-PUFA from *A. limacinum* BCC52274 and *Mortierella sp.* is a more suitable and sustainable sources of LC-PUFA for aquaculture feed, as it improves the growth and immunity in *L. vannamei* PL.

## Supporting information

**S1 Table. Proportions of dried AL and ARACO used to supplement *Artemia* for the feed experiment.**
(DOCX)

**S2 Table. Primer sequences and PCR conditions for the qPCR analysis of shrimp immune genes in *L. vannamei*.**
(DOCX)

**S3 Table. Fatty acid compositions and total fatty acid (TFA) contents (% dry weight) of *Artemia* supplemented with different ratios of DHA:ARA.**
(DOCX)

**S1 File. Fatty acid composition in AL and ARASCO oil.**
(XLSX)

**S2 File. Fatty acid composition in *Artemia* from each feed group.**
(XLSX)

**S3 File. Fatty acid composition in PL1 and PL18 from each feed group.**
(XLSX)

**S4 File. Growth performance analysis of PL18 fed with enriched *Artemia* in different feed groups.**
(XLSX)

**S5 File. Transcription levels of immune genes in PL18 fed with *Artemia* with varying ratios of n-3: n-6 PUFA supplementation.**
(XLSX)

**S6 File. The UPLC-HRMS/MS analysis of eicosanoids in PL samples fed with enriched *Artemia*.**
(XLSX)

**S7 File. Effects of pathogenic infection in PLs fed with enriched *Artemia*.**
(XLSX)

## Acknowledgments

We sincerely thank DSM Nutritional Products (Thailand) Ltd. for their generous provision of ARASCO and alpha-tocopherol used in this study. We thank the members of the Advanced Diagnostics and Biomarker Discovery Research Team at BIOTEC for their technical support during sample collection.

## Author contributions

**Conceptualization:** Sage Chaiyapechara, Waraporn Jangsutthivorawat, Metavee Phromson, Wananit Wimuttisuk.

**Formal analysis:** Virak Visudtiphole, Panida Unagul, Waraporn Jangsutthivorawat, Metavee Phromson, Pacharawan Deenarn, Punsa Tobwor, Pisut Yotbuntueng, Surasak Jiemsup, Wananit Wimuttisuk.

**Funding acquisition:** Virak Visudtiphole, Wananit Wimuttisuk.

**Investigation:** Virak Visudtiphole, Panida Unagul, Sage Chaiyapechara, Waraporn Jangsutthivorawat, Metavee Phromson, Siriporn Tala, Pacharawan Deenarn, Punsa Tobwor, Pisut Yotbuntueng, Surasak Jiemsup, Suganya Yongkiettrakul, Looksorn Koichai, Wananit Wimuttisuk.

**Methodology:** Virak Visudtiphole, Panida Unagul, Sage Chaiyapechara, Waraporn Jangsutthivorawat, Metavee Phromson, Pisut Yotbuntueng, Surasak Jiemsup, Suganya Yongkiettrakul, Looksorn Koichai.

**Project administration:** Virak Visudtiphole, Wananit Wimuttisuk.

**Resources:** Virak Visudtiphole, Wananit Wimuttisuk.

**Supervision:** Virak Visudtiphole, Panida Unagul, Wananit Wimuttisuk.

**Validation:** Virak Visudtiphole, Pisut Yotbuntueng, Wananit Wimuttisuk.

**Visualization:** Virak Visudtiphole, Pisut Yotbuntueng, Wananit Wimuttisuk.

**Writing – original draft:** Virak Visudtiphole, Wananit Wimuttisuk.

**Writing – review & editing:** Virak Visudtiphole, Sage Chaiyapechara, Pacharawan Deenarn, Punsa Tobwor, Pisut Yotbuntueng, Suganya Yongkiettrakul, Wananit Wimuttisuk.

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
