## [Decision Letter · Decision Letter 0]

Dear Dr. Wimuttisuk,

Thank you for submitting your manuscript to PLOS ONE. After careful consideration, we feel that it has merit but does not fully meet PLOS ONE’s publication criteria as it currently stands. Therefore, we invite you to submit a revised version of the manuscript that addresses the points raised during the review process.

We look forward to receiving your revised manuscript.

Kind regards,

Lee Seong

Academic Editor

PLOS ONE

Journal Requirements:

“This research has received funding support from the Functional Ingredient & Food Innovation Program, National Center for Genetic Engineering and Biotechnology [grant number P19-51807] to VV and the NSRF via the Program Management Unit for Human Resources & Institutional Development, Research and Innovation [grant number B05F640184] to WW.”

Reviewers' comments:

Reviewer's Responses to Questions

**Comments to the Author**

1. Is the manuscript technically sound, and do the data support the conclusions?

Reviewer #1: Yes

Reviewer #2: Yes

2. Has the statistical analysis been performed appropriately and rigorously?

Reviewer #1: Yes

Reviewer #2: Yes

3. Have the authors made all data underlying the findings in their manuscript fully available?

Reviewer #1: Yes

Reviewer #2: Yes

4. Is the manuscript presented in an intelligible fashion and written in standard English?

Reviewer #1: Yes

Reviewer #2: Yes

Reviewer #1: Dear Author

Thank you for your good manuscript I accept it only with minor change

please add new references and revised again discussion .and short material and methods and explain more result and add main idea

Reviewer #2: This study provides valuable insights into sustainable alternatives to fish oil in aquaculture feed by evaluating the efficacy of Aurantiochytrium limacinum BCC52274 (AL) and Mortierella sp. oil (ARASCO) as sources of long-chain polyunsaturated fatty acids (LC-PUFAs) for Pacific white shrimp (Litopenaeus vannamei) post-larvae. The experimental design, which involved Artemia enrichment with varying DHA:ARA ratios, effectively demonstrated that different LC-PUFA profiles can be strategically applied to promote either growth or immune enhancement in shrimp. Notably, Group A (DHA:ARA 100:0) achieved the highest biomass and body weight, while Group D (DHA:ARA 25:75) exhibited elevated prophenoloxidase II expression and increased production of anti-inflammatory eicosanoids. Furthermore, several enriched groups showed a reduced viral load following WSSV infection, highlighting the immunoprotective potential of these dietary supplements. Overall, the findings support the viability of AL and ARASCO as sustainable, functional replacements for traditional marine-derived fatty acids in shrimp aquaculture. The manuscript is well-written and well-structured, with only a few minor comments for consideration.

Minor Comments:

1. Lines 211–217: Please specify the sample size and the age of the post-larvae used for RNA extraction.

2. The concentration of RNA used for cDNA synthesis should be mentioned.

3. Vibrio harveyi is not mentioned in the abstract and should be included.

4. The discussion section would benefit from a deeper analysis of the findings in the context of existing literature, or by proposing hypotheses where data are currently lacking.

**Do you want your identity to be public for this peer review?** For information about this choice, including consent withdrawal, please see our Privacy Policy

Reviewer #1: No

Reviewer #2: No

---

## [Author Response · Author response to Decision Letter 1]

14 Jul 2025

Answer: We have provided a rebuttal letter (this document), a marked-up copy of our manuscript, and an unmarked version of the revised manuscript based on the journal’s instructions.

2. Answer: We have made clarification regarding the financial disclosure in the second-to-last paragraph of the updated cover letter (yellow highlight). Should you have any more questions, please let us know.

Journal Requirements:

Answer: We have adjusted the manuscript so that it meets PLOS ONE’s style requirements in the revised version of the manuscript.

“This research has received funding support from the Functional Ingredient & Food Innovation Program, National Center for Genetic Engineering and Biotechnology [grant number P19-51807] to VV and the NSRF via the Program Management Unit for Human Resources & Institutional Development, Research and Innovation [grant number B05F640184] to WW.”

Answer: We have added the suggested sentence regarding the roles of the funders into the updated cover letter. Please change the online submission form on our behalf.

Answer: We have rechecked all references by going through the journal’s database, google scholars, as well as databases for the retracted publications. However, we were unable to identify the retracted paper.

Reviewer Comments to the Author

Reviewer #1: Dear Author

Thank you for your good manuscript I accept it only with minor change

please add new references and revised again discussion .and short material and methods and explain more result and add main idea

Answer: We thank Reviewer #1 for your kind suggestions. As there was no specific instruction given in the reviewers’ response by PLOS ONE (which reference to add and which part of the discussion to be revised), we have improved our manuscript as follows:

1) Added new references can be found in line 416-419, 440-449, 479-481, and 496-498.

2) Revised the discussions: The discussion was revised to correct grammatical errors as well as the added discussion on the main idea (line 424-435) as well as more discussion based on the added new references shown in 1).

3) Short materials and methods: We have shortened the materials and methods as follows.

3.1) Under the section pathogen challenge, we have removed “to assess the effect of Artemia enrichment on the immunological phenotype of the PL in response to bacterial and viral infection, PL19 were challenged with Vibrio harveyi or white spot syndrome virus (WSSV). Details of the challenge are as follows."

3.2) Under the Quantitative real-time PCR analysis to detect WSSV copy number, we have removed the following sentences.

“The water volume in each tank was adjusted from 150 L to 180 L on day 13 of the experiment to account for the larger size of the shrimp.”

“As WSSV infection in PLs typically does not result in mortality [33,34], the effect of feed supplementation on the WSSV-infected PLs was evaluated by determining the WSSV copy number using a qPCR analysis.”

“The extracted DNA was spectrophotometrically quantified (OD260) and qualified (OD260/OD280 ~ 1.6-1.9)”

and “Oligonucleotide primers were obtained from Integrated DNA Technologies in South Korea.”

4) Explain more results: We have added an explanation in the result section in line 280-282.

5) Added the main idea: A proposed model was added in line 410 to 435 as well as the added Conclusion section (line 507-518)

Reviewer #2: This study provides valuable insights into sustainable alternatives to fish oil in aquaculture feed by evaluating the efficacy of Aurantiochytrium limacinum BCC52274 (AL) and Mortierella sp. oil (ARASCO) as sources of long-chain polyunsaturated fatty acids (LC-PUFAs) for Pacific white shrimp (Litopenaeus vannamei) post-larvae. The experimental design, which involved Artemia enrichment with varying DHA:ARA ratios, effectively demonstrated that different LC-PUFA profiles can be strategically applied to promote either growth or immune enhancement in shrimp. Notably, Group A (DHA:ARA 100:0) achieved the highest biomass and body weight, while Group D (DHA:ARA 25:75) exhibited elevated prophenoloxidase II expression and increased production of anti-inflammatory eicosanoids. Furthermore, several enriched groups showed a reduced viral load following WSSV infection, highlighting the immunoprotective potential of these dietary supplements. Overall, the findings support the viability of AL and ARASCO as sustainable, functional replacements for traditional marine-derived fatty acids in shrimp aquaculture. The manuscript is well-written and well-structured, with only a few minor comments for consideration.

Answer: We thank the reviewer for your kind comments. We have clarified the manuscript based on the comments provided in the following section.

Minor Comments:

1. Lines 211–217: Please specify the sample size and the age of the post-larvae used for RNA extraction.

Answer: PL18 (N = 30) were used for RNA extraction to determine the impact of DHA:ARA supplementation on shrimp immune genes. We have added this information to line 217. For the consistency of the manuscript, we have also added the information regarding the PL sample size in each rearing tank (N = 15,000) in line 139-140 and the number of PL used to determine WSSV copy number (N = 35) in line 173.

2. The concentration of RNA used for cDNA synthesis should be mentioned.

Answer: The author has added the concentration of RNA used for the cDNA synthesis in lines 222 along with the amounts of cDNA used in each real-time PCR reaction in lines 229.

3. Vibrio harveyi is not mentioned in the abstract and should be included.

Answer: The authors have included a sentence regarding the V. harveyi experiment in the abstract (lines 48-49).

4. The discussion section would benefit from a deeper analysis of the findings in the context of existing literature, or by proposing hypotheses where data are currently lacking.

Answer: We have added a proposed model and more discussion in line 419-431, 490-496, and 500-511.

---

## [Editor Report · Decision Letter 1]

Effects of microbial-derived long-chain polyunsaturated fatty acids from Aurantiochytrium limacinum BCC52274 and Mortierella sp. on growth and immunity in Litopenaeus vannamei post-larvae

PONE-D-25-31399R1

Dear Dr. Wimuttisuk,

We’re pleased to inform you that your manuscript has been judged scientifically suitable for publication and will be formally accepted for publication once it meets all outstanding technical requirements.

Kind regards,

Lee Seong

Academic Editor

PLOS ONE